# Merging Models with Fisher-Weighted Averaging

**Michael Matena**      **Colin Raffel**
Department of Computer Science
University of North Carolina at Chapel Hill
{mmatena,craffel}@cs.unc.edu

## Abstract

Averaging the parameters of models that have the same architecture and initialization can provide a means of combining their respective capabilities. In this paper, we take the perspective that this "merging" operation can be seen as choosing parameters that approximately maximize the joint likelihood of the posteriors of the models' parameters. Computing a simple average of the models' parameters therefore corresponds to making an isotropic Gaussian approximation to their posteriors. We develop an alternative merging procedure based on the Laplace approximation where we approximate each model's posterior as a Gaussian distribution whose precision matrix corresponds to its Fisher information. We first show that our "Fisher merging" technique provides a performance boost in settings where simple parameter averaging is currently used – specifically, robust fine-tuning and model ensembling. Then, we compare merging to standard gradient-based transfer learning and demonstrate that merging enables a fundamentally different method for transferring capabilities across models. Specifically, we show that Fisher merging is competitive with gradient-based transfer learning approaches (while being significantly cheaper) in intermediate-task training and domain-adaptive pre-training. We also show that our merging procedure makes it possible to combine models in previously unexplored ways. We release our code to facilitate future research into methods for merging models.[1]

## 1   Introduction

How should we transfer knowledge and capabilities across trained models? One popular approach is transfer learning [44], which fine-tunes a pre-trained model on a target task through additional gradient-based training. The preparatory step of pre-training the model on a data-rich task ideally instills useful "knowledge" into the network's parameters, which allows the model to learn more rapidly and effectively when fine-tuned on a downstream task of interest. Transfer learning has therefore become a particularly important and omnipresent tool across many fields, including natural language processing [57, 13, 9, 52, 53, 46] and computer vision [43, 24, 68]. Recently, it has been shown that training on an "intermediate" task between pre-training and fine-tuning can further boost performance through additional transfer of capabilities from the intermediate task [47, 60, 51, 48]. Alternatively, continued self-supervised training on unlabeled domain-specialized data can serve as a form of domain adaptation [19].

All of the aforementioned transfer learning methods transfer knowledge by using a trained network to initialize another network followed by iterative gradient descent. While demonstrably powerful, several drawbacks arise from this: First, improvements to ancestor models cannot be passed down to descendants; instead, we must restart the whole process from the improved ancestor model, throwing away our previous work. For example, if we fine-tune a pre-trained model on a downstream task,

---

[1] https://github.com/mmatena/model_merging

36th Conference on Neural Information Processing Systems (NeurIPS 2022).

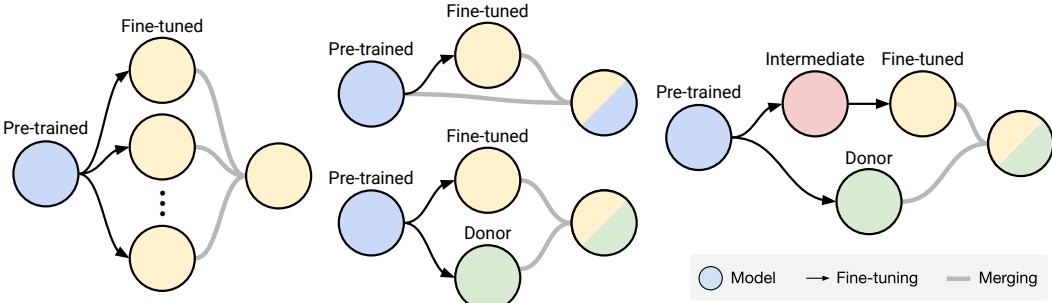

Figure 1: Merging patterns considered in this work. *Left:* Merging many fine-tuned models as a form of ensembling. *Center, top:* "Robust fine-tuning" [66] , where a fine-tuned model is merged with the pre-trained model to improve performance on the original pre-training task. *Center, bottom:* Merging a fine-tuned model with a "donor" task, analogous to intermediate-task transfer learning [47, 51]. *Right:* Merging an intermediate-task trained model with a donor model.

but then the pre-trained model is improved through additional training, we must re-fine-tune the new model on our downstream task if we want to confer benefits from this additional pre-training. Furthermore, if we gain access to a checkpoint that has been fine-tuned on a useful intermediate task, we must again throw away our previous work and fine-tune from the intermediate task checkpoint. Existing methods for transfer learning also have the disadvantage of only being able to transfer information from a single model. While it may be possible to train on multiple intermediate tasks sequentially, one quickly either runs into a combinatorial explosion of saved checkpoints or faces the issue of "catastrophic forgetting" in continual learning [28]. In addition to slowing down experimentation by preventing reuse of work, these drawbacks impose limitations on the types of transfer that can occur.

A less common way of transferring capabilities across models is to simply average their parameters. This procedure, which we call "merging", is generally only feasible when the models being averaged share a common architecture and initialization. Merging is the core component of the FedAvg algorithm used in Federated Learning [39], where updates to a shared model computed by individual workers that are training on different datasets are combined by simpling averaging the updates. Recently, Wortsman et al. [66] demonstrated that merging can be used to improve robustness to domain shift in fine-tuned models by averaging the parameters of the original pre-trained model with the fine-tuned parameters. Merging is also a common way of performing ensembling [49, 67], where the parameters of individual models trained on the same dataset are averaged to create a single performant model.

In this work, we view model merging as approximately maximizing the joint likelihood of the models' posterior distribution over parameters. Since gradient-based maximum likelihood training only provides a point estimate of the posterior, some approximation of the posterior distribution is required. When an isotropic Gaussian distribution is used to approximate the posterior (with identity precision matrix and mean set to the model's parameter values), we show that maximizing the joint likelihood across models is equivalent to simply averaging their parameters. We therefore refer to merging models by averaging parameters as *isotropic merging*. The view of merging as maximizing the joint likelihood of model posteriors suggests that using a better estimate of the posterior distribution may yield improved merging results. This leads us to introduce *Fisher merging*, which leverages the Laplace approximation by using the diagonal of each model's Fisher information as the precision matrix for that model's posterior.

Empirically, we demonstrate that merging models with Fisher merging outperforms isotropic merging in a variety of settings. We first focus on the existing applications of model ensembling [49, 67] and improving fine-tuned model robustness [66]. Then, we demonstrate for the first time that merging is a viable alternative to traditional gradient-based transfer learning. Specifically, we compare merging to intermediate-task transfer learning [47, 51] and domain-adaptive pre-training [19], finding that merging can achieve comparable performance at significantly lower cost. Additionally, we show that merging can provide an additional boost to models created via traditional intermediate-task training. This provides a concrete example of transfer that is fast and easy with merging but onerous

or impossible to do with existing methods. Diagrams of the merging patterns we consider in this work are shown in fig. 1.

The rest of our paper is structured as follows: In section 2, we provide necessary background and detail our Fisher merging procedure. Section 3 provides experimental results on model ensembling, robust fine-tuning, intermediate-task training, and domain adaptation. We explore related works in section 4 and provide conclusions and thoughts on future work in section 5.

## 2   Weighted Parameter Averaging for Model Merging

Our focus is on procedures for *model merging*, i.e. averaging the parameters of models that share an architecture and initialization. In this section, we first frame the common practice of averaging together model parameters as approximately maximizing the joint likelihood of model posteriors. Specifically, we show that parameter averaging corresponds to using an isotropic Gaussian as the approximate posterior for each model. We then introduce *Fisher merging*, which uses the model's diagonal Fisher information matrix as the precision matrix of the Gaussian approximate posterior. Fisher merging can be implemented by setting each merged parameter value to a weighted average of the corresponding parameter values from the original models, with the weighting for each parameter determined by its Fisher information. In addition, we add model-level weightings as additional hyperparameters to set the relative importance of each model.

### 2.1   Isotropic merging

Consider the problem setting where we have $M$ trained neural networks with parameters $\theta_1, \ldots, \theta_M$ and our goal is to create a single neural network with parameters $\theta$ that, loosely speaking, inherits the capabilities of the $M$ trained neural networks. Assume that all of these neural networks share a common architecture and had the same set of initial parameter values before being trained. *Merging* attacks this problem by finding the parameters $\theta$ that maximize the joint likelihood of the posterior distributions of the $M$ models. Unfortunately, typical neural network training procedures do not provide access to a posterior distribution, which necessitates approximation. If the posterior of each model is approximated via an isotropic Gaussian with mean set to the model's parameters, the optimization problem can be written as $\theta^* = \text{argmax}_\theta \sum_i \log p(\theta|\theta_i, I)$ where $p(\theta|\theta_i, I)$ is the probability distribution of the aforementioned approximate isotropic Gaussian posterior distribution used for model $i$ and $I$ is the identity matrix. This optimization problem has a closed-form solution given by $\theta^* = \frac{1}{M} \sum_i \theta_i$, i.e. an average of the model parameters. Such an averaging procedure has been used in past work aiming to combine model capabilities, e.g. in federated learning [39], model ensembling [49, 67], and robust fine-tuning [66].

### 2.2   Per-model weights

In this work, we additionally introduce model-specific scalar hyperparameters $\lambda_i, i \in \{1, \ldots, M\}$ into the model merging framework described above. Specifically, we change the optimization problem to $\theta^* = \text{argmax}_\theta \sum_i \lambda_i \log p(\theta|\theta_i, I)$ where $\lambda_i \geq 0, \sum_i \lambda_i = 1$. In the case of isotropic merging, this changes the solution to $\theta^* = \sum_i \lambda_i \theta_i$, These hyperparameters provide control over the importance assigned to each of the models that are being merged. For example, when using merging to perform ensembling we might expect each model to be equally important and therefore set $\lambda_i = 1/M$ for all $i$. On the other hand, when mimicking the setup of intermediate-task training where the capabilities of a "donor" model are used to improve performance of a recipient model, we might weigh the recipient model more highly. Wortsman et al. [66] introduce a similar hyperparameter $\alpha$ when averaging the parameters of two models and report results for varying values of $\alpha$.

### 2.3   Laplace Approximation

Framing merging as approximate maximization of the joint posterior likelihood reveals that simple parameter averaging is implicitly using an isotropic Gaussian posterior approximation. Such an approximation may be overly simplistic and lead to degraded performance. To explore improved merging procedures, we consider improved methods for creating an approximate posterior from a point estimate. Specifically, we use the Laplace approximation to the posterior, which corresponds to a second-order Taylor expansion of the log density around a mode [36, 10]. This leads to a Gaussian

approximation $\mathcal{N}(\theta, H^{-1})$ of the posterior, where $H$ is the Hessian matrix and $\theta$ are the model's trained parameter values. More precisely, we assume that the parameter values $\theta$ of a trained neural network are a local maximum of the posterior. It can then be shown that the precision matrix of the Laplace approximation is given by the Fisher information matrix of the network at $\theta$.

The Fisher information matrix $F_\theta$ [16, 3] of a neural network $p_\theta(y|x)$ trained to predict an output $y$ from input data $x$ is a $|\theta| \times |\theta|$ positive semidefinite matrix given by the formula

$$F_\theta = \mathbb{E}_x \left[ \mathop{\mathbb{E}}_{y \sim p_\theta(y|x)} \nabla_\theta \log p_\theta(y|x) \nabla_\theta \log p_\theta(y|x)^T \right]. \tag{1}$$

It can be shown that the Fisher information matrix coincides with the Hessian $H$ at modes of the distribution [45], explaining its use in the Laplace approximation. The Fisher information matrix $F_\theta$ can also be used to relate changes in the model parameters to changes in the model output by noting that $\mathbb{E}_x [D_{\mathrm{KL}}(p_\theta(y|x)||p_{\theta+\delta}(y|x))] \approx \frac{1}{2}\delta^T F_\theta \delta$ as $\delta \to 0$, where $D_{\mathrm{KL}}$ denotes the KL-divergence [45].

As the full Fisher matrix takes $O(|\theta|^2)$ memory to store, it quickly becomes impractical for all but the smallest models. We are thus forced to use an approximation to the full Fisher in practice. In this paper, we follow the common practice of using the diagonal of the Fisher matrix [28]. While other methods (e.g. [1]) exist for estimating the Fisher, we leave their exploration for future work. In our experiments, we estimated the diagonal of the Fisher matrix via

$$\hat{F}_\theta = \frac{1}{N} \sum_{i=1}^{N} \mathop{\mathbb{E}}_{y \sim p_\theta(y|x_i)} (\nabla_\theta \log p_\theta(y|x_i))^2, \tag{2}$$

where $x_1, \ldots, x_N$ are drawn *i.i.d.* from the dataset that was used to train the model. The expectation over $y$ can be estimated via sampling from $p_\theta(y|x_i)$ or computed exactly when the number of classes is small. We note that computing the Fisher requires $N$ per-example gradients, which can be straightforwardly computed for neural networks using backpropagation. This makes computing the diagonal Fisher have roughly the same computational cost as training on $N$ examples.

## 2.4 Fisher Merging

Having noted that the Laplace approximation provides a tractable way to obtain a better approximation to the posterior, we now use it to create an improved merging procedure that we call *Fisher merging*. Letting $F_1, \ldots, F_M$ correspond to the diagonal approximate Fisher matrices, we construct $p(\theta|\theta_i, F_i)$ as a Gaussian-distributed posterior over the parameters of the merged model with mean $\theta_i$ and precision $F_i$. To obtain the merged model, we find a single set of parameters that is given a high probability under all posteriors. Formally, we have

$$\theta^* = \mathrm{argmax}_\theta \sum_{i=1}^{M} \lambda_i \log p(\theta|\theta_i, F_i), \tag{3}$$

which has the closed-form solution

$$\theta^{*(j)} = \frac{\sum_{i=1}^{M} \lambda_i F_i^{(j)} \theta_i^{(j)}}{\sum_{i=1}^{M} \lambda_i F_i^{(j)}}, \tag{4}$$

where $j = 1, \ldots, |\theta|$. Intuitively, we can think of Fisher merging as computing a weighted average of the parameter values in each model where the weighting is done according to each parameter's Fisher information. Since the Fisher information is a local property of a single parameter value, Fisher merging might be less performant when applied to models whose parameters are far apart in parameter space. We therefore limit our focus to models that were trained from the same initialization.

**Numerical Issues.** Note that (4) can run into numerical issues when the Fisher is close to zero across all models for a given parameter. In practice, we choose a privileged "target model" in all of our experiments and "default" to the parameter's value in the target model in these cases. An alternative would be to take an average weighted only by the merging coefficients (i.e., pretend the Fisher is the same across all models). In practice, the choice of a "default" value for these parameters had little impact on performance (likely because a small Fisher value implies that changing the parameter has a minute effect on the model's outputs and is therefore relatively unimportant to the model's behavior).

**Unmergeable Parameters.** In many cases, we have some parameters from each model that do not appear in all of the models we are merging. For example, this includes having task-specific classification heads on top of a common body architecture. We handle this by only applying the merging procedure (3) to the shared body parameters and keeping the task-specific heads unchanged. Although this may lead to a distribution shift in the classification head inputs, we found it to work well in practice for the datasets and tasks we consider.

# 3 Experiments

Our first experimental goal is to validate that our use of an improved estimate of the posterior yields improved merging performance. To test this hypothesis, we apply Fisher merging to two settings where isotropic merging has already proven successful: Model ensembling [49, 67] and robust fine-tuning [66]. Then, we demonstrate that Fisher merging provides a cheap and effective alternative to traditional transfer learning pipelines by validating its performance in intermediate-task transfer learning [47, 51] and domain-adaptive pre-training [19]. Finally, we demonstrate that merging opens up new paths of transferring capabilities across models by demonstrating a boost in performance when merging an intermediate task-trained model with different donor models.

## 3.1 Ensembling

An existing application of isotropic merging is for *ensembling*, i.e. combining models trained on the same dataset to obtain better predictions. Ensembling is most commonly performed by averaging the predictions of the individual models. This form of ensembling requires computing the output of all $M$ models in the ensemble, thereby increasing the computational cost by a factor of $M$ compared to computing the output for a single model. A cheaper alternative is to average the parameters of the models themselves. This approach is diagrammed in fig. 1, left. Such an approach is used in the classical method of Polyak averaging [49], where parameter values from the final $M$ iterations of training are averaged. More recently, Wortsman et al. [67] introduced the "Model Soup" approach where fine-tuned models with different hyperparameter settings are averaged to improve performance. To the best of our knowledge, all parameter-averaging ensemble methods have used isotropic merging, i.e. an unweighted average.

To test whether Fisher merging provides a boost over isotropic merging when averaging parameters for ensembling, we consider ensembling fine-tuned checkpoints derived from the same pre-trained model. Specifically, we consider the BERT-Base model [13] fine-tuned on the RTE [8], MRPC [14], and SST-2 [59] datasets. For each dataset, we use five fine-tuned checkpoints downloaded from the Hugging Face model hub.[2] These checkpoints were fine-tuned with a variety of hyperparameter settings that were not chosen by us, so our experimental setting most closely matches the "Model Soup" approach [67]. A list of the checkpoints used is available in appendix A. Since we do not anticipate that any member of the ensemble should be given a larger weight, we set $\lambda_i = 1/5$ for all models.

Our results are shown in fig. 2. We report validation set scores for Fisher merging, isotropic merging, and prediction ensembling (specifically, averaging the output probabilties of all models). Fisher merging significantly outperforms isotropic merging in all cases and attains comparable performance to prediction ensembling. Notably, performing inference after merging is $M\times$ cheaper than prediction ensembling, suggesting that merging can provide a cheaper alternative to standard ensembling procedures.

## 3.2 Robust Fine-Tuning

Recently, Wortsman et al. [66] found that while fine-tuning a pre-trained vision model tends to improve performance on the downstream task, it also tends to decreases accuracy on the original pre-training task. They therefore propose a "robust fine-tuning" procedure called WiSE-FT that computes a weighted average of the original pre-trained parameters and the fine-tuned parameters. Different weighting values produce different trade-offs between pre-training and fine-tuning task performance. In some cases, robust fine-tuning can even improve performance on the original pre-training task without sacrificing performance on the downstream fine-tuning task relative to traditional fine-tuning.

---

[2]https://huggingface.co/models

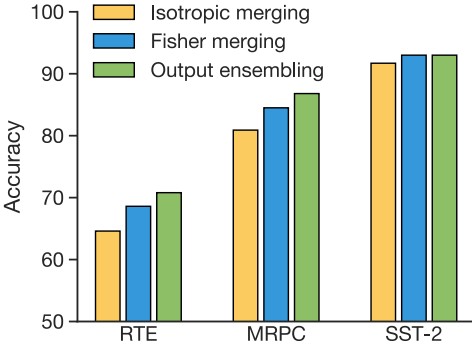

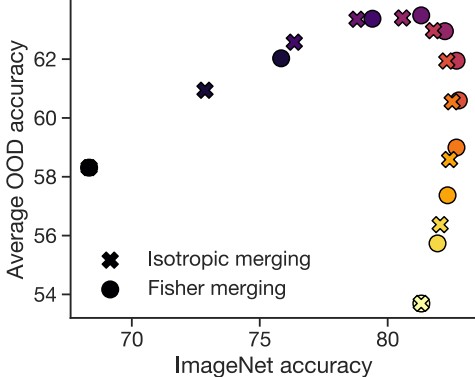

Figure 2: Validation set accuracy for ensembles of five fine-tuned BERT models using different ensembling methods on the RTE, MRPC, and SST-2 datasets. Fisher merging produces a single model that performs comparably to output ensembling while being $5\times$ cheaper.

Figure 3: IID (ImageNet) and average out-of-domain (OOD) accuracy across five OOD datasets when using the WiSE-FT procedure [66] with either Fisher or isotropic merging. Dark to light color indicates increasing $\lambda_1$ from 0 to 1.

This procedure implicitly uses isotropic merging and therefore provides another natural testbed for determining whether Fisher merging provides a boost in performance. A schematic of robust fine-tuning is shown in fig. 1, center top.

We use the codebase and experimental setup of Wortsman et al. [66] exactly, simply replacing isotropic merging with Fisher merging. For full details of this setup, we refer to Wortsman et al. [66]. As a short summary, we apply WiSE-FT to the ImageNet [11, 58] pre-trained ViT-B/16 model [15] on five out-of-domain (OOD) datasets: ImageNet-A [21], ImageNet-R [20], ImageNet Sketch [62], ImageNet V2 [56], and ObjectNet [4]. Following Wortsman et al. [66], we measure IID (ImageNet) and OOD performance when averaging together the original pre-trained model parameters and parameters from models fine-tuned on each of the OOD datasets, varying $\lambda_1$ (the averaging weight for the pre-trained model, called $\alpha$ by Wortsman et al. [66]) from 0 to 1 in 0.1-step increments (with $\lambda_2 = 1 - \lambda_1$ correspondingly decreasing from 1 to 0). To determine whether Fisher merging confers a boost in performance, we compare parameter averaging using either isotropic or Fisher merging.

We plot the IID (ImageNet) accuracy against the average accuracy on the five OOD datasets for varying values of $\lambda_1$ in fig. 3, with plots for individual OOD datasets in fig. 7 (appendix). Fisher merging produces a significantly better trade-off between IID and OOD accuracy. In particular, Fisher merging seems to general improve IID accuracy compared to isotropic merging. For example, for the value of $\lambda_1$ producing the best average OOD accuracy, Fisher merging produces about 1% higher IID accuracy than isotropic merging.

### 3.3 Intermediate-task training

Having established that Fisher merging produces better results than isotropic merging in settings where merging has been attempted before, we now explore the use of merging as an alternative to a gradient-based transfer learning procedure. Specifically, we explore intermediate-task training [47, 51], where a model is fine-tuned on an intermediate "donor" task before being trained on the target task of interest. To the best of our knowledge, no prior work has considered parameter averaging as a way of performing intermediate-task transfer learning. For the most part, intermediate-task training has mainly been considered in the NLP domain; as such, we limit our experiments to the BERT [13] and RoBERTa [33] pre-trained language models. To enable comparison to past work, we mostly explored merging pairs of models but we are interested in exploring merging more than two models in future work. As in section 3.1, we made use of fine-tuned BERT and RoBERTa checkpoints from the Hugging Face repository [65].

Following previous work [47, 51], we first ran experiments using BERT-base on the GLUE benchmark [61]. The GLUE benchmark consists of the sentence acceptability task CoLA [64], the sentiment detection task SST-2 [59], the paraphrase detection tasks MRPC and QQP [14, 23], the sentence similarity task STS-B [7], and the natural language inference (NLI) tasks MNLI, QNLI, RTE, and

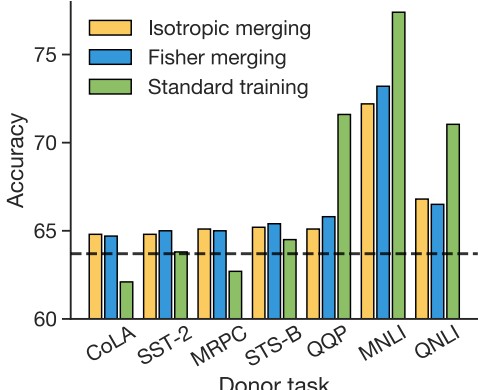

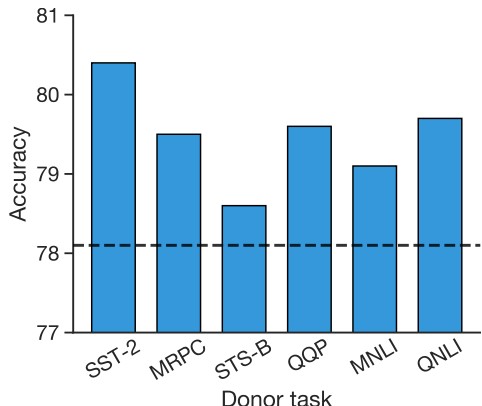

Figure 4: Validation set accuracy on RTE when performing intermediate-task training with datasets from GLUE as the donor task. Dashed line denotes RTE accuracy without intermediate-task training.

Figure 5: Validation accuracy on RTE after first fine-tuning on MNLI, then fine-tuning on RTE, and finally Fisher merging with various donor task models. Dashed line denotes RTE accuracy after MNLI intermediate-task training.

WNLI [6, 54, 8, 31]. All of the GLUE tasks are classification tasks except for STS-B, which is a regression task with a score ranging from 0 to 5. To simplify computation of the Fisher, we turn STS-B into a classification task by partitioning the continuous label into 25 equally-sized buckets [53]. Following common practice, we do not run experiments on WNLI due to the tricks required to get a good score [12, 29]. See Wang et al. [61] for more details on these tasks and their associated metrics. We detail how we obtained fine-tuned checkpoints on these tasks in appendix C. We computed a diagonal Fisher approximation for each checkpoint using up to 4096 examples from the corresponding train set. Since it is not clear a priori what weighting coefficients $\lambda_i$ to use in this setting, we chose $\lambda_i$ by a grid search with 50 points, using the score on the first 2048 validation examples as the selection metric. We compare Fisher merging to isotropic merging as well as a standard gradient-based intermediate-task fine-tuning baseline [47]. A diagram of intermediate-task merging is shown in fig. 1, center bottom.

In initial experiments (reported in tables A1 to A3), we performed intermediate-task training for possible pair of datasets from the GLUE benchmark. Congruent with past work [47, 51, 60], we found that intermediate-task training provided the most notable performance boost when the RTE dataset was the target. We therefore focus on RTE results in the main text. Figure 4 shows the results of intermediate-task training of BERT-base with RTE as the target task and the other GLUE datasets as donor tasks, using Fisher merging, isotropic merging, or standard gradient-based training. Notably, performing gradient-based intermediate-task training *hurts* on some datasets, whereas merging always helps. Fisher merging gets comparable or better performance than isotropic merging with the largest gap observed when using MNLI as the intermediate task. On the other hand, merging performs worse than standard gradient-based training when using MNLI as the donor task.

**Exploring new paths for transfer** Given this performance gap, we were interested to see whether merging could provide an additional boost on top of gradient-based intermediate-task training. We therefore performed Fisher merging on a BERT-base model that was first fine-tuned on MNLI and then fine-tuned on RTE. A diagram of this setup is shown in fig. 1, right. This procedure does not have a direct analog in traditional gradient-based, and as we will show later, performing multi-stage gradient-based intermediate-task training generally harms results.

We consider Fisher merging the intermediate-task trained RTE model with all GLUE tasks and show the results in fig. 5. Fisher merging provides a boost over gradient-based intermediate-task training for all tasks. Interestingly, a boost is still conferred when merging with an MNLI-trained model, suggesting that merging provides a complementary path for transferring capabilities across models.

**Scaling to RoBERTa-large** Seeing that merging can provide a boost on top of intermediate-task training, we explored whether this boost could still be obtained for a stronger model than BERT-base. We therefore applied the same procedure to a RoBERTa-large RTE model that had been

fine-tuned from an MNLI intermediate checkpoint. Our donor models were the original RoBERTa-large checkpoint (i.e., not fine-tuned on MNLI) fine-tuned on MRPC, RTE, STS-B, and SST-2. We additionally ran a sequential gradient-based fine-tuning baseline where we started with the MNLI checkpoint, fine-tuned on the donor task, and then fine-tuned on the target task.

The results are shown in fig. 6. We find merging provides a boost in performance even on the more performant RoBERTa model. The largest boost of 2.2 points came from Fisher merging with another RTE checkpoint, which is reminiscent of using merging for ensembling. Notably, including an additional intermediate task in gradient-based training significantly harmed performance compared to performing intermediate-task training on MNLI alone. We hypothesize this is related to the phenomena of catastrophic forgetting [17], where the model's capabilities on MNLI are forgotten as it is trained on the next intermediate task. Nevertheless, this illustrates model merging's ability to sidestep the issue of catastrophic forgetting and enable exploration of novel transfer strategies.

**Costs**    We had previously noted that our merging procedure could potentially be substantially more efficient than standard gradient-based fine-tuning. To measure this claim concretely, we computed the FLOPs required for fine-tuning and merging an RTE checkpoint based on the heuristics described in Kaplan et al. [26]. Fine-tuning BERT-base on RTE for 10 epochs would require about 5.5e14 FLOPs. Our merging procedures require computing the merged checkpoint (eq. (4)) and then evaluating it on the validation set with Fisher merging also requiring the estimation of the Fisher matrix (eq. (2)) beforehand. These steps require about 4.0e8, 2.0e12, and 9.1e13 FLOPs respectively, resulting in a roughly $6\times$ lower total cost compared to fine-tuning for Fisher merging and $275\times$ lower cost for isotropic merging. We note that the Fisher matrix only needs to be computed once and can be reused for subsequent merges, which amortizes the most expensive step in Fisher merging.

To explore methods for further reducing costs, we experimented with using fewer examples to estimate the Fisher. Specifically, we experimented with intermediate-task Fisher merging of BERT-base with MNLI as the donor task and RTE as the target task. The results are shown in table A4. While using the full training set to estimate the Fisher produced the best performance (73.4%), using only 256 examples to estimate the Fisher only produced a mild degradation in accuracy (72.7%) and still outperformed the isotropic merging baseline. This suggests that computing the Fisher over fewer examples could further reduce computational costs without sacrificing a great deal of accuracy.

### 3.4    Domain Adaptation

We now turn our attention to the "domain-adaptive pre-training" (DAPT) approach for domain adaptation advocated by Gururangan et al. [19], which is methodologically similar to intermediate-task training. DAPT consists of additional pre-training of an original general-purpose pre-trained checkpoint on domain-specific unlabeled data. We explore the benefits of merging in an experimental setup similar to Gururangan et al. [19]. We focus on the biomedical (BIOMED) and computer science (CS) domains because they correspond to the classification tasks that saw the largest gains from domain-adaptive pre-training in [19]. Namely, we experimented with the CHEMPROT [30] relation classification task on the BIOMED domain. On the CS domain, we used the citation intent task of ACL-ARC [25] and the relation classification task of SCIERC [35]. Following Gururangan et al. [19], we report macro-$F_1$ for ACL-ARC and SCIERC, and we report micro-$F_1$ for CHEMPROT. We used RoBERTa-base [33] as our baseline model. Appendix D includes full details of the pre-training, fine-tuning, and merging procedures used.

We present our results in table 1. Merging provided the largest boost on ACL-ARC, and outperformed traditional fine-tuning in this setting. We only observed a minor improvement in performance on CHEMPROT and SCIERC. We note that our boosts from gradient-based fine-tuning were smaller than reported in [19], which was likely because we were only able to train on public data and we applied domain-adaptive pre-training for fewer steps. However, our results are consistent in the sense that ACL-ARC received the largest boost and CHEMPROT received the smallest boost.

## 4    Related Work

Like our work, elastic weight consolidation (EWC) [28] uses the Laplace approximation to the posterior over model parameters to create a regularizer to prevent catastrophic forgetting in the context of continual learning. While their framework supports the use of posteriors from multiple

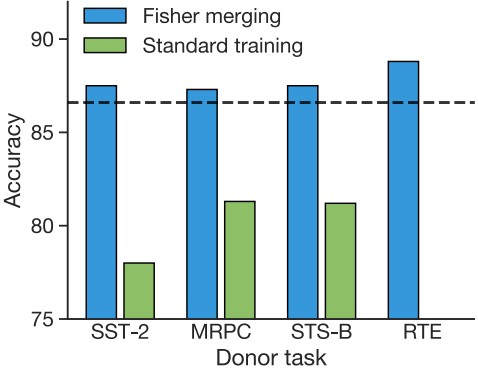

Figure 6: Validation accuracy on RTE using the setup of fig. 5, but with RoBERTa-large instead of BERT-base. "Standard training" fine-tunes on MNLI, then the donor task, then RTE. Dashed line denotes MNLI intermediate-task training.

| Method | ChemProt | ACL-ARC | SciERC |
|---|---|---|---|
| Unmerged | $82.7_{0.3}$ | $70.5_{3.2}$ | $81.0_{0.4}$ |
| Fisher | $83.1_{0.4}$ | $73.2_{1.7}$ | $81.3_{0.5}$ |
| Isotropic | $82.8_{0.4}$ | $72.5_{2.3}$ | $81.7_{0.5}$ |
| Fine-tuned | $82.5_{0.1}$ | $71.5_{3.0}$ | $81.6_{1.0}$ |

Table 1: Domain adaptation results. "Unmerged" refers to checkpoints fine-tuned from RoBERTa-base. "Fisher" and "Isotropic" refer to the result of merging those checkpoints with the domain-adaptive pre-trained (DAPT) checkpoint. "Fine-tuned" refers to models fine-tuned from the DAPT checkpoint. Subscripts provide the standard deviation across five trials.

models as well, they restrict such models to be previous checkpoints of a continually trained model. EWC keeps the model from losing previously acquired knowledge while merging provides a means of directly adding new knowledge to a model.

Some other existing procedures such as distillation [22] and ensembling [42] can also be thought of as combining or transferring knowledge between neural networks. However, those methods represent knowledge solely through the output of models. The knowledge contained within the parameters of a network will necessarily be greater than the knowledge contained in its output [2]. Hence, methods that directly combine model parameters such as merging have the potential to be more powerful than those methods. Furthermore, our merging procedure has an efficient and closed-form solution (eq. (4)) while distillation requires iterative gradient descent-based training.

Isotropic checkpoint averaging is used by federated learning [39] and Polyak averaging [49]. However, the checkpoints merged by those methods can be thought of coming from the same training run of single model. We believe we are the first to demonstrate cross-task transfer coming from checkpoint averaging and to explore it in the context of transfer learning. However, adapting ideas from federated learning such as [32, 63] could provide a fruitful avenue for future model merging research.

Natural gradient descent refers to an optimization procedure that uses KL-divergence of model predictions as a distance measure rather than the Euclidean distance in parameter space employed by regular gradient descent [3]. It does this by performing gradient descent on a Riemannian manifold with the Fisher information matrix as its metric [45]. In practice, this amounts to using the Fisher as a preconditioner during gradient descent. Some work on natural gradient descent may prove relevant for model merging such as using Kronecker-factorized Fisher matrices as an alternative to the diagonal approximation employed in this paper [37, 18, 38]. More broadly, in the field of information geometry the Fisher information matrix plays the role of a metric on a Riemannian manifold [40]. This has led to explorations of model averaging using tools from information geometry, e.g. [5, 41, 50].

## 5   Conclusion

In this paper, we introduced *Fisher merging*, a way to combine the capabilities of different models by computing a weighted average of their parameters. Fisher merging is motivated by a novel perspective of parameter averaging as maximizing the joint likelihood of model posteriors. Through extensive experiments, we demonstrated that using the Fisher information as a weight on the contribution of each parameter outperforms using an unweighted average. Furthermore, we showed that Fisher merging attains comparable and sometimes better performance than traditional gradient-based transfer learning methods at significantly lower costs. Our experiments also demonstrated various merging strategies that would be onerous with traditional gradient-based training, which opens up new avenues for transferring capabilities across models. In future work, we plan to investigate different methods for approximating the Fisher information and model posteriors as well as more esoteric combinations of models.

## Acknowledgments and Disclosure of Funding

This work was supported by the NSF CAREER award under Grant No. 2145822.

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
