# OpenReview forum: "Merging Models with Fisher-Weighted Averaging"
_NeurIPS.cc/2022/Conference — NeurIPS 2022 Accept_

### Official Review · Reviewer_dofC · 2022-07-03

**Rating:** 7
**Confidence:** 5
**Soundness:** 3 good
**Presentation:** 4 excellent
**Contribution:** 4 excellent

**Summary:**

This paper explore "merging" models, i.e. combining the parameters of two or more models into a single model. It advances an increasing popular research direction in averaging parameters of models that share the same architecture and initialization. Their main contribution is to go beyond the way parameters are typically averaged, by having individual weights for each of the parameters. For such, the authors advocate for finding a model with the most likely set of parameters, considering the posteriors of all models being averaged. Their method is based on approximating each model's posterior as a Gaussian distribution whose precision matrix corresponds to its Fisher information. This then leads to a closed-form solution for which weights to use for each parameter when averaging multiple models. Authors additionally have global hyper-parameters which can be tuned to control the importance of each model when averaging. The authors present a series of experiments in both natural language processing and computer vision, contrasting their Fisher-weighted averaging with previous work where simple averaging is used (i.e., without any parameter-specific weights). The authors also explore new settings for combining models, in addition to existing ways like robust fine-tuning or model soups. Code  for replicating the experiments will be publicly released.
Overall, this paper presents a solid advance in our understanding and methods for combining models. Considering that pre-training is becoming increasingly expensive and intractable to repeat, this line of work is very timely. I believe this paper would be of interest to many in the community, and I recommend its acceptance.

**Questions:**

Have the authors explored larger language models? I believe this line of work is most attractive when pre-training is prohibitively expensive (if pre-training is cheap, why not just combine the datasets and pre-train again?). It would also be interesting to show that this can work with parameter-efficient fine-tuning methods like adapters.

**Limitations:**

The authors discuss some limitations of their method and how to address them (for instance, numerical issues and unmergeable parameters). As previously mentioned in the Strengths and Weaknesses section, I believe better understanding of when Fisher merging provides large improvements over Isotropic merging would greatly strengthen the paper.

**Strengths And Weaknesses:**

**Strengths**

1) The research direction this paper explores is timely, and I believe would be of interest to many.

2) The ideas presented in this work are novel, and it departs significantly from previous work.

3) The experiments presented are comprehensive and solid. Compared to standard (isotropic) merging, Fisher merging provides improvements in most cases, sometimes by large amounts. Moreover, their method does not incur in any extra compute at inference time compared to the models being merged. Finally, the fact that the the authors conduct experiments both in natural language processing and computer vision showcases the generality of their findings.

4) I found the experiments presented in Section 3.1 to be particularly interesting, since authors combine models with external checkpoints, created independently by multiple people. Considering model zoos like the Hugging Face model hub are growingly rapidly, this experiment is a great proof of concept of a future were we take better advantage of such resources.

5) The paper provides a great review of related work, and contextualizes itself and its contributions well with respect to recent literature.

**Weaknesses**

1) In some cases, the improvements gained via Fisher merging (compared to Isotropic merging, which has been previously explored) are small or even inexistent. Better understanding *when* Fisher merging provides larger gains compared to Isotropic merging would greatly strengthen the paper.

2) Compared to Isotropic merging, Fisher merging can require extra compute. In section 3.3, the authors claim that merging is more computationally efficient than fine-tuning, but this only holds true when assuming that all checkpoints being merged are already available. While this can be true for many datasets, there are scenarios where it does not hold, for instance, for recently released, less popular or private tasks. Moreover, the proposed procedure might be infeasible in some cases: for instance, consider fine-tuning a CLIP model on ImageNet, as in Sec. 3.2. CLIP models are public, but their training data is not. Given that the authors advocate for using data from the dataset used to train the model for computing the Fisher matrix (L. 141), how can this be done if it is not public? If other datasets can be used for computing the Fisher matrix, which ones to chose? Does the choice matter? Further exploring and discussing this would strengthen the paper.

---

> ### Author Response · Authors · 2022-08-02
> **Author response**
>
> Thank you for your thorough review. We are glad you appreciated the timeliness of our submission; indeed, the existence of model zoos and availability of many pre-trained models was a major motivator of our work. We have responded to your comments and questions below.
>
> > In some cases, the improvements gained via Fisher merging (compared to Isotropic merging, which has been previously explored) are small or even inexistent. Better understanding when Fisher merging provides larger gains compared to Isotropic merging would greatly strengthen the paper.
>
> In our experiments, we found that Fisher merging always matched or exceeded the performance of isotropic merging. Their performance was similar in cases where merging conferred limited benefits (e.g. in intermediate-task training experiments where the donor task provided no real improvement in accuracy). We have added additional discussion of these cases to the paper to help readers understand when and why Fisher merging can provide gains.
>
> > Compared to Isotropic merging, Fisher merging can require extra compute.
>
> While this is true, we find in §3.3 under “Costs” that using only 256 examples to estimate the Fisher can produce strong performance. This implies that computing the Fisher can be made as costly as training on a few batches of data, which is easily justifiable given that Fisher merging typically improves performance.
>
> > Given that the authors advocate for using data from the dataset used to train the model for computing the Fisher matrix (L. 141), how can this be done if it is not public? If other datasets can be used for computing the Fisher matrix, which ones to chose? Does the choice matter?
>
> In this paper, we assume that the data used to train the model is available during merging. This is true for CLIP, since we consider only CLIP after was fine-tuned on ImageNet or other “OOD” datasets. In general, we think this is a reasonable assumption since most public models were trained on public data, and most private models would only be merged by someone who both had access to the private model and the data it was trained on. We agree that it would be interesting to explore estimating the Fisher on other datasets in future work.
>
> > Have the authors explored larger language models? I believe this line of work is most attractive when pre-training is prohibitively expensive (if pre-training is cheap, why not just combine the datasets and pre-train again?). It would also be interesting to show that this can work with parameter-efficient fine-tuning methods like adapters.
>
> We agree that Fisher merging could be especially helpful with giant models. However, we lack the computational resources to experiment with such models and therefore focus on moderately large models (i.e. hundreds of millions of parameters).

---

> > ### Comment · Reviewer_dofC · 2022-08-03
> > **Thank you for the response**
> >
> > I appreciate your time to think about my comments and suggestions. I stick with my pre-rebuttal rating as I have already recommended acceptance.

---

### Official Review · Reviewer_PTzq · 2022-07-11

**Rating:** 6
**Confidence:** 4
**Soundness:** 3 good
**Presentation:** 3 good
**Contribution:** 2 fair

**Summary:**

The paper studies the problem of how to effectively average existing deep learning models, which has broad application background in transfer learning. The authors proposed to derive a **geometric centroid** of the given models based on information geometry. Based on certain simplifications (such as diagonal Fisher information matrix, FIM), the resulting model is in closed form based on the FIM in eq.(4). On real-world transfer learning experiments, the authors showed that the proposed method is better than empirical averaging.

**Questions:**

Sec. 2. related expressions regarding the posterior can be better formulated and clearly written. For example, why $\theta^{\star}=\sum_{i}\log{}p(\theta|\theta_i,I)$? and why $p(\theta|\theta_i,I)$ is the posterior? What is the associated prior? These formulations can be improved.

The presented work is a simple application of Fisher information into modern machines. Most of the cited papers are from recent deep learning communities. The idea of using information geometry for model averaging is not new. Related literature where similar ideas were known and used should be better acknowledged.  For example, see  [1][2][3].

[1] "Specification Uncertainty and Model Averaging", Larry M. Bartels, 1997.

[2] "Sided and Symmetrized Bregman Centroids", Nielsen & Nock, 2009.

[3] "Model Selection and Model Averaging in Phylogenetics: Advantages of Akaike Information Criterion and Bayesian Approaches Over Likelihood Ratio Tests", David Posada, Thomas R. Buckley, 2004.

There should be more discussion on why it works better than the empirical averaging approach. Instead, the authors introduced the method "as is" without sufficient theoretical arguments or motivation.

**Limitations:**

The paper is theoretical and does not have a negative societal impact as far as I can tell.

**Strengths And Weaknesses:**

Pros:

- The proposed method is efficient based on a diagonal approximation of the Fisher matrix (as in Adam).
- Strong experimental results with extensive study in modern deep learning models.

Cons:

- The paper can be regarded as an application of information geometry into deep learning. The theoretical depth can be improved to explain why it works well.
- Some historical literature on model selection and information geometry can be more carefully investigated.

---

> ### Author Response · Authors · 2022-08-02
> **Author response**
>
> Thanks for your comments and suggestions. We’ve responded to each below and updated our paper draft accordingly.
>
> > The paper can be regarded as an application of information geometry into deep learning. The theoretical depth can be improved to explain why it works well… There should be more discussion on why it works better than the empirical averaging approach. Instead, the authors introduced the method "as is" without sufficient theoretical arguments or motivation.
>
> Thank you for this suggestion. We agree that our paper can be viewed as an application of information theory into deep learning. Our method falls out of the perspective of model averaging as approximate joint posterior maximization combined with the use of the Laplace method. Given that the Laplace method is well-studied and is by definition a better posterior estimate than an isotropic Gaussian, we chose not to focus the paper on additional theoretical depth, but instead focused on showing the effectiveness of our method in a variety of settings. If there are specific theoretical details/findings that you think would be beneficial to include, let us know and we can aim to do so.
>
> > Sec. 2. related expressions regarding the posterior can be better formulated and clearly written.
>
> Thank you for noting this. We updated the text surrounding the equations you mentioned to provide a better explanation.
>
> > Some historical literature on model selection and information geometry can be more carefully investigated… The presented work is a simple application of Fisher information into modern machines. Most of the cited papers are from recent deep learning communities. The idea of using information geometry for model averaging is not new. Related literature where similar ideas were known and used should be better acknowledged. For example, see [1][2][3].
>
> Thank you for suggesting this interesting related work. We have added the suggested references to our updated draft.

---

### Official Review · Reviewer_Dr1z · 2022-07-11

**Rating:** 7
**Confidence:** 4
**Soundness:** 3 good
**Presentation:** 3 good
**Contribution:** 3 good

**Summary:**

The paper proposes a novel method for merging (pre-trained) models with the same architecture and initialisation set up. The merging strategy for obtaining the new model based on the previous ones is based on the Laplace approximation, and particularly, on the Fisher information matrix. In the manuscript, the advantages of merging via Fisher information matrix is illustrated within simpler strategies based on isotropic Gaussian approximations or just output ensembles. Experimental results using (very) large models -- i.e. BERT, show a significant performance while at the same time being simple and x5-x6 cheaper than other approaches.

**Questions:**

Some additional questions that I would appreciate to be answered.

- *Q1.* If I understand correctly, once each $m$-th model is trained, we are able to compute the Fished matrix in a *posthoc* manner. Once F
 Is computed/approximated, we can use it for training the new model merging. Does this last merging re-visit old data? Or F is assumed to be fixed and known without the need of re-accessing $x$ or $y$?

- *Q2.* This is just a curiosity, but as far as I know, BERT models use self-attention layers in their architecture. Do these type of layers/operations introduce unmergeable parameters? In other recent Laplace papers, some type of layers could not be considered due to the difficulties for computing Hessians.

- *Q3.* How many models are being merged in the experiments? And how large is the space of parameters? Could the authors provide some numbers on the validation set accuracy of the individual pre-trained models?

- *Q4.* Are convolutional layers considered also as parameters to be merged?


**Limitations:**

Yes

**Strengths And Weaknesses:**

**Strengths.**

The idea of merging models to transfer capabilities and knowledge to new models without the need of exhaustive re-training from the same data is sound and novel. The paper positively illustrates the problem considered and related work is well reviewed in my opinion. While the methodology section could be slightly improved (see weaknesses), the experimental section is strong, mainly due to the consideration of big models or other neural network (NN) based architectures in the results. While being simple, performance seems promising. In general, I had not seen results on merging such large models like BERT or relative ones. Also details on FLOPs or computational cost saved from merging is a good point for the community.

**Weaknesses.**
In my opinion, there are two small points of weakness:

- *Section 2:* The methodology described in Sec. 2 for the isotropic + Fisher merging could be improved, or at least connect better with probabilistic modelling. For me, was not difficult to follow and key details are included, but perhaps some intermediate details or equations are missing. For instance, equation in L102 introduces densities $p(\theta|\theta_i, I)$, not being 100% clear enough where they come from. Similarly, in L129-L132, Fisher matrix is introduced within data $x,y$ which are not previously defined.

- *Laplace Approximation:* There is a recent interest in the community on Laplace approximation and Hessian-based methods, mainly due to their utility for providing alternatives to Bayesian NNs (approximating posterior on the parameters). Even being a different goal, I find the paper somehow connected, and some references could be perhaps included (Laplace Redux, + previous and follow-up approaches). Additionally, I am a bit surprised about the lack of difficulties for computing the Fisher information matrix (even being diagonalised) in such enormous architectures.. I would appreciate if authors could elaborate a bit more on this and the computational cost.

---

> ### Author Response · Authors · 2022-08-02
> **Author response**
>
> Thank you for your detailed review. We have addressed your suggestions below and in our updated draft.
>
> > The methodology described in Sec. 2 for the isotropic + Fisher merging could be improved, or at least connect better with probabilistic modelling.
>
> Thank you for these recommendations. We have edited the passages you noted to make them more clear. $p(\theta | \theta_i, I)$ is the probability density of an approximate Gaussian posterior of a model with parameters $\theta_i$, and $x$ and $y$ are placeholder variables for the input data and predicted output of the neural network.
>
> > Laplace Approximation: There is a recent interest in the community on Laplace approximation and Hessian-based methods, mainly due to their utility for providing alternatives to Bayesian NNs (approximating posterior on the parameters). Even being a different goal, I find the paper somehow connected, and some references could be perhaps included (Laplace Redux, + previous and follow-up approaches).
>
> We agree that this is an important connection. In our paper, we intentionally focused on the Laplace approximation as it provides a very simple and general means of approximating the posterior in cases where only a point estimate is available (as is the case in the vast majority of trained neural networks). We have added a call for future work using improved methods for posterior approximation as well as a reference to the Laplace Redux paper.
>
> > I am a bit surprised about the lack of difficulties for computing the Fisher information matrix (even being diagonalised) in such enormous architectures.. I would appreciate if authors could elaborate a bit more on this and the computational cost.
>
> Computing the Fisher’s diagonal can be done simply and cheaply by leveraging per-example gradient calculations, which can be computed for neural networks via backpropagation. Estimating the diagonal Fisher on $N$ examples therefore has roughly the same computational cost as training on $N$ examples. Since a single Fisher value is compute for each parameter, it incurs the same storage cost as the model itself. We have added this information to our paper.
>
> > Once F Is computed/approximated, we can use it for training the new model merging. Does this last merging re-visit old data? Or F is assumed to be fixed and known without the need of re-accessing $x$ or $y$?
>
> Yes, F is assumed to be fixed and known, which allows us to re-use existing Fisher matrices when performing additional merges.
>
> > This is just a curiosity, but as far as I know, BERT models use self-attention layers in their architecture. Do these type of layers/operations introduce unmergeable parameters? In other recent Laplace papers, some type of layers could not be considered due to the difficulties for computing Hessians.
>
> No, there is no problem merging the parameters of self-attention, since self-attention is computed via a series of simple matrix-vector products.
>
> > Q3. How many models are being merged in the experiments?
>
> In all experiments in the paper, we merged two models.
>
> > And how large is the space of parameters?
>
> The size of the models range from 86M parameters (CLIP ViT-B/16) to 354M parameters (RoBERTa-large)
>
> > Could the authors provide some numbers on the validation set accuracy of the individual pre-trained models?
>
> For ensembling experiments, we used existing models from the Hugging Face model hub, which are enumerated in Appendix A. The corresponding pages on the Hugging Face model hub give their individual validation set accuracies before ensembling. For robust fine-tuning experiments, the validation set accuracy on ImageNet (IID) or the OOD dataset(s) can be seen in the $\lambda_1 = 1$ and $\lambda_1 = 0$ points of the plot, respectively. For intermediate-task training, the pre-trained model scores are shown as a dashed line in each figure. For domain adaptation, the pre-trained model scores are shown in the “Unmerged” row in Table 1.
>
> > Q4. Are convolutional layers considered also as parameters to be merged?
>
> Yes, convolutional layers would be readily mergeable.

---

> > ### Comment · Reviewer_Dr1z · 2022-08-09
> > **Rebuttal Acknowledgment**
> >
> > Thanks to the authors for such a good response to my questions. To be honest, I liked very much the paper while reading it, but considering the points indicated now in the rebuttal, I perceive that it is even better. Particularly, I remark that if the computation of the Fisher matrix is that cheap, no data is revisited (as we can re-use other Fisher matrices) and the framework works for ~400M of parameters, this is a great work. To me, the only detail is that only two models are merged in the experimental results paper, and this could be seen as a tiny weakness. Perhaps merging more models would be a great contribution in a future submission. Finally, as no questions remain to me, I update my score accordingly.

---

### Official Review · Reviewer_c1b6 · 2022-07-11

**Rating:** 5
**Confidence:** 4
**Soundness:** 2 fair
**Presentation:** 3 good
**Contribution:** 2 fair

**Summary:**

This paper proposes a method for casting the problem of averaging parameters from multiple models with shared architecture as seeking a MAP estimate of the joint posterior with Laplace approximation for individual trained models.
Empirical experiments on natural language processing datasets are presented, showing its effectiveness on model ensembling, fine-tuning, and domain adaptation tasks.

**Questions:**

- For neural networks with batch normalization, and spectral normalization, how would the parameters be averaged in the proposed method?
- How would the parameter uncertainty be addressed in the proposed method?

**Limitations:**

Yes.

**Strengths And Weaknesses:**


#### Strengths
- The method proposed in this work is conceptually simple and intuitive. The paper is generally well-written and easy to follow.
- The empirical evaluation covers a wide range of tasks and metrics, including both task-specific performance and computational efficiency.

#### Weaknesses
- While I understand the point that the authors aim to propose an alternative to the plain parameter averaging approach, the introduced method is still seeking a point estimation for the joint posterior which is based on the Laplace approximation. Thus the issues of the Laplace approximation would also exist in the proposed approach, and it is hard to tell how the uncertainty in the final estimation is addressed given the results included.
- Quite a few experimental results are presented, however, only on natural language datasets and tasks. As a general parameter averaging approach, it would be beneficial to consider other data modalities as well.
- As discussed in the first point about the limitations of the Laplace approximation, for efficient estimation of the Fisher matrix, diagonal approximation is made which leaves the structure of the task ignored.

---

> ### Author Response · Authors · 2022-08-02
> **Author response**
>
> Thanks for your thorough review and suggestions. We’ve addressed each of your points below.
>
> > the issues of the Laplace approximation would also exist in the proposed approach, and it is hard to tell how the uncertainty in the final estimation is addressed given the results included.
>
> We agree that the Laplace approximation can be a crude estimate of the posterior. However, we were interested in a simple method that could be applied to existing trained neural networks (where only a point estimate of the posterior is available). Merging does not explicitly require a detailed characterization of the uncertainty - yielding a new point estimate provides the same capabilities as the original models being merged. Future work could explore whether using improved methods of posterior approximation produces better merging results. We added a note to the paper to highlight this.
>
> > Quite a few experimental results are presented, however, only on natural language datasets and tasks. As a general parameter averaging approach, it would be beneficial to consider other data modalities as well.
>
> In our paper, we consider two main settings: Fine-tuning pre-trained language models (§3.1, §3.3, §3.4) and adapting an ImageNet classifier to new domains/datasets (§3.2). The latter setting does not deal with natural language but rather image classification. We believe including these two settings covers a large proportion of popular applications of transfer learning.
>
> > As discussed in the first point about the limitations of the Laplace approximation, for efficient estimation of the Fisher matrix, diagonal approximation is made which leaves the structure of the task ignored.
>
> We agree that an improved estimate of the Fisher matrix could yield improved merging performance. However, in this paper we were focused on demonstrating that Fisher merging provided a merging technique that was competitive or better than traditional gradient-based training while being more efficient. We note that in future work we are interested in exploring the benefit of using improved estimates of the Fisher.
>
> > For neural networks with batch normalization, and spectral normalization, how would the parameters be averaged in the proposed method?
>
> For models that use moving average estimates in their parameterization (e.g. the moving average estimate of first and second moments used in batch normalization), we could envision many possibilities: First, these parameters could be averaged (without Fisher weighting, as they do not have well-defined gradients); second, they could be used selectively depending on the input data (e.g. if models trained on different tasks/domains were merged, the parameters corresponding to the task could be used); third, they could be re-estimated based on the statistics of data during inference; and finally, they could be set to the values from the target task in intermediate-task experiments. We are optimistic that these strategies would work well given that we also encountered “unmergeable” parameters in the form of classifier layers (§2.4, “Umergeable Parameters”). We note that such parameters are uncommon in modern networks like those in our paper (which typically use layer normalization).
>
> > How would the parameter uncertainty be addressed in the proposed method?
>
> Since our method operates on point estimates and produces a point estimate, it does not make use of or provide a notion of parameter uncertainty. However, we anticipate that leveraging models/methods that provide a better approximation of the posterior could improve merging. We have added a note to our paper to emphasize this for future work.

---

### Official Review · Reviewer_XQHE · 2022-07-11

**Rating:** 5
**Confidence:** 4
**Soundness:** 3 good
**Presentation:** 3 good
**Contribution:** 3 good

**Summary:**

Transfer learning is ubiquitous in deep learning methods, and gradient-based transfer learning is usually dominant in these areas where a pre-trained network in fine-tuned for a task. The issue with this approach is that after fine-tuning the network becomes better at the current task at hand but might 'forget' the previous task. To alleviate this issue the authors (along with others) suggest taking the 'average' of the two networks (or more networks if other fine-tuned networks are available). This approach combines the strengths of multiple models such that they retain their performance on their respective individual tasks. The authors extend this approach by considering the weighted mean of the different models (at an individual parameter level) and guide the weighing of the parameters using their respective information matrices.

**Questions:**

Can the following two sentences be clarified a bit more?

- "Since the Fisher information is a local property of a single parameter value, we limit our focus to models that were trained from the same initialization." Do you refer to "same pre-trained model" being fine-tuned multiple times? Why is this relevant if the parameter spaces are exactly the same?

- "a small Fisher value ultimately means that the parameter is relatively unimportant to the model’s behavior." I will guess that a small information will imply that the parameter cannot be estimated reliably from the data, and thus, can be removed from the "averaging". Is this a suitable interpretation?

Can Figure 3 be elaborated a bit more:

- Can you show the transition of regularization values (perhaps using different transparency).
- What are the values that simultaneously

Figure 4 shows that isotropic merging and Fisher merging perform similar to each other on an average, and standard fine-tuning can outperform these approaches on some occasions. Can
the authors provide some guidelines on when one of these three approaches should be considered?
~

**Limitations:**

Some limitation of the method, e.g., numerical issues have been addressed in the paper. It will be great to address situations where Fisher merging performs better than isotropic merging and vice versa.

**Strengths And Weaknesses:**

Strengths
- The paper tackles an interesting and relevant problem.
- The paper presents significant empirical evidence.

Weaknesses
- The idea presented is straightforward and the main contribution of the paper is empirical results corroborating the proposed framework, and thus, lacks theoretical arguments.
- Some intuitions and sentences in the paper are not obvious.
- The arguments made in the paper are a bit difficult to follow.

---

> ### Author Response · Authors · 2022-08-02
> **Author response**
>
> Thanks for the helpful suggestions. We’ve responded to each of your concerns and questions below and in our updated draft.
>
> > The idea presented is straightforward and the main contribution of the paper is empirical results corroborating the proposed framework, and thus, lacks theoretical arguments.
>
> While we do not prove any new theory pertaining to our approach, we believe that our perspective of viewing parameter averaging as approximately maximizing the joint likelihood of the posteriors of the models' parameters is new and useful. This theoretically-sound perspective not only leads to better methods for parameter averaging (like our Fisher merging approach), but should also prompt future work where parameter averaging is used (e.g. ensembling, federated averaging, etc.).
>
> > Some intuitions and sentences in the paper are not obvious… The arguments made in the paper are a bit difficult to follow.
>
> Thank you for noting this. If you have any specific places where things are unclear beyond those that you identified under “Questions”, we would be happy to update our paper to make the intuition and arguments easier to follow.
>
> > "Since the Fisher information is a local property of a single parameter value, we limit our focus to models that were trained from the same initialization." Do you refer to "same pre-trained model" being fine-tuned multiple times? Why is this relevant if the parameter spaces are exactly the same?
>
> If we are averaging two models whose parameters are far apart in parameter space, we lose the ability to interpret parameter averaging as approximately maximizing posteriors because it makes assumptions that are local properties of the parameter values. As such, we only average models that are trained from the same initial parameter values (e.g. fine-tuning from the same pre-trained checkpoint). We don’t consider cases where the model was fine-tuned multiple times in the paper. The fact that the parameterization is the same doesn’t help here - what matters is the distance in parameter space, rather than the dimensionality of the space itself. We have clarified this sentence in our updated draft.
>
> > "a small Fisher value ultimately means that the parameter is relatively unimportant to the model’s behavior." I will guess that a small information will imply that the parameter cannot be estimated reliably from the data, and thus, can be removed from the "averaging". Is this a suitable interpretation?
>
> If the Fisher information of a given parameter is small, it means that changing that parameter has little effect on the model’s predictions. For such a parameter, it could simply be that the model does not need to make use of that degree of freedom in order to perform well. We have updated this sentence to clarify.
>
> > Can Figure 3 be elaborated a bit more
>
> Thanks, as you suggested we have updated the figure to make the color of each point correspond to its averaging value.
>
> > Figure 4 shows that isotropic merging and Fisher merging perform similar to each other on an average, and standard fine-tuning can outperform these approaches on some occasions. Can the authors provide some guidelines on when one of these three approaches should be considered? … It will be great to address situations where Fisher merging performs better than isotropic merging and vice versa.
>
> Thank you for pointing out that a recap of our experimental findings would be beneficial. In all experimental settings, we found that Fisher merging almost always performed better than Isotropic merging; they typically tied in cases where merging (or fine-tuning) provided limited benefit. We did find in the intermediate-task training experiments that fine-tuning could outperform Fisher merging, but Fisher merging incurs minimal additional costs assuming fine-tuned models are available (§3.3, “Costs”). Merging also outperformed gradient-based training in our domain adaptation experiments (§3.4) and also enables new ways of transferring capabilities across models (§3.3, “Exploring new paths for transfer”).

---

### Author Response · Authors · 2022-08-02
**Updated submission**

Thanks very much to all of the reviewers for their constructive suggestions. We have responded to all reviewer comments and questions below and have updated our draft accordingly. Changes made include:

- Clarified notation in Section 2
- Added references for recent uses of the Laplace approximation and information geometry-based model averaging
- Provided more detail about how the Fisher diagonal can be computed efficiently
- Clarified why we assume that all models to be merged are fine-tuned from the same initial parameter values
- Added more intuition about why parameters with low Fisher information do not pose numerical issues
- Updated Figure 3 to add coloring that makes it clear what values of $\lambda_1$ each point corresponds to

If the reviewers have additional suggestions, we would be happy to incorporate them into an updated draft. Thanks again for your input.

---

### Meta-Review · Area_Chair_ZAdS · 2022-08-26

**Recommendation:** Accept
**Confidence:** Certain

**Metareview:**

Most reviewers agree that the paper proposes some interesting novel ideas and  that its
strengths overcome some of its weaknesses (eg lack on theoretical guarantees). As such, we are
think that the paper is worth publishing and expect  the authors to improve the manuscript
in accordance with some of the reviewers comment.


**Award:**

No

---

### Decision · Program_Chairs · 2022-09-14

Accept